# GILT: An LLM-Free, Tuning-Free Graph Foundational Model for In-Context Learning

## Abstract

Graph Neural Networks (GNNs) are powerful tools for precessing relational data but often struggle to generalize to unseen graphs, giving rise to the development of Graph Foundational Models (GFMs). However, current GFMs are challenged by the extreme heterogeneity of graph data, where each graph can possess a unique feature space, label set, and topology. To address this, two main paradigms have emerged. The first leverages Large Language Models (LLMs), but is fundamentally text-dependent, thus struggles to handle the numerical features in vast graphs. The second pre-trains a structure-based model, but the adaptation to new tasks typically requires a costly, per-graph tuning stage, creating a critical efficiency bottleneck. In this work, we move beyond these limitations and introduce **G**raph **I**n-context **L**earning **T**ransformer (GILT), a framework built on an LLM-free and tuning-free architecture. GILT introduces a novel token-based framework for in-context learning (ICL) on graphs, reframing classification tasks spanning node, edge and graph levels in a unified framework. This mechanism is the key to handling heterogeneity, as it is designed to operate on generic numerical features. Further, its ability to understand class semantics dynamically from the context enables tuning-free adaptation. Comprehensive experiments show that GILT achieves stronger few-shot performance with significantly less time than LLM-based or tuning-based baselines, validating the effectiveness of our approach.

## 1 Introduction

Graph Neural Networks (GNNs) have emerged as the standard for processing graph data, achieving state-of-the-art performance on a wide range of single-graph tasks (Kipf & Welling, 2017; 2016; Xu et al., 2019). However, their fundamental limitation is a lack of generalization: a GNN trained on one graph often fails to transfer to an unseen graph with different features or topology (Hu et al., 2020b). In parallel, the artificial intelligence area has been reshaped by the success of foundation models that exhibit remarkable transferability in domains like language (Brown et al., 2020) and vision (Radford et al., 2021). This confluence of GNNs' limitations and the power of the foundation model paradigm has spurred intense interest in a new frontier: the Graph Foundational Model (GFM).

However, the extreme heterogeneity in graph data presents a fundamental obstacle to realizing the vision of GFM. Unlike text or images, which benefit from a universal vocabulary such as tokens or pixels, graphs lack this common foundation. Each graph can possess its own unique and arbitrary feature space, with varying dimensions and semantics; a distinct target space with its own unique structure like discrete classes or continuous values; and a vastly different topological structure (Mao et al., 2024). Consequently, the parameters of a conventional GNN architecture are fundamentally tied to the specific feature and output formats of its training data, making the model inherently non-transferable (Hu et al., 2020b). This core challenge of bridging graph heterogeneity has driven the development of the main GFM paradigms to date (Liu et al., 2025).

The first primary paradigm tackles the heterogeneity problem by leveraging Large Language Models (LLMs) to create a unified semantic space (Li et al., 2024a; Fan et al., 2024; Ren et al., 2024). The core strategy involves applying an LLM to interpret the textual information associated with a graph's nodes and edges, mapping diverse features, even labels, into a shared space. Although highly effective for text-rich graphs like citation networks, this approach introduces a dependency on textual data (Zhao et al., 2025; 2024a; Sun et al., 2025a; Yu et al., 2025a). Consequently, it is

unsuitable for graphs with numerical, categorical, or purely structural data, as is common in fields like molecular biology (Wu et al., 2017).

The second major paradigm, known as graph prompting, takes a more direct and graph-native solution to heterogeneity by real-time parameter adaptation, typically involving pre-training a GNN encoder on large-scale data and then adapting it to downstream tasks (Sun et al., 2023a; 2025a; Yu et al., 2025a). While this approach avoids the text-dependency issue, it introduces a dependency on tuning. The need to modify model weights for each new graph or task persists (Sun et al., 2023b), creating a significant efficiency bottleneck and diverging from the promise of a truly "out-of-the-box" foundational model (Zhao et al., 2025).

In this work, we move beyond the two barriers by introducing the **Graph In-context Learning Transformer (GILT)**, a framework designed to be both **LLM-free and tuning-free**. Our key innovation is to reframe few-shot graph tasks, spanning node, edge, and graph classification, as a unified token-based in-context learning problem. GILT's architecture first uses a graph-native pipeline to tokenize a task, converting its structure and features into a standardized set of contextual tokens. These tokens are then processed by a specialized Transformer that learns the task's semantics directly from the prompted examples. This in-context learning mechanism allows GILT to dynamically interpret even unseen feature and label spaces at inference time, completely bypassing the need for textual information or parameter updates.

We empirically validate GILT on a diverse suite of benchmarks spanning node, link, and graph classification, with results confirming its state-of-the-art few-shot performance. Its LLM-free design allows it to operate directly on text-independent graphs with numerical or structural features, where text-based models are often either inapplicable or require laborious pre-processing to manually create textual descriptions. Concurrently, its tuning-free nature provides a significant efficiency advantage, making it orders of magnitude faster than methods that require per-graph gradient updates and positioning it as a more practical and scalable solution. Our code is available at this anonymous link: `https://anonymous.4open.science/r/inductnode-313E/`.

Our main contributions in this work are as follows:

- We design and implement GILT, a LLM-free, tuning-free In-Context Learning architecture. reframing few-shot graph problems as a universal token-reasoning task.
- We collect a diverse suite of datasets for pretraining and trained one model for multiple tasks through a graph-native tokenization pipeline and a specialized two-stage ICL Transformer.
- We provide comprehensive empirical validation establishing GILT's superiority. Our experiments demonstrate its state-of-the-art few-shot performance in text-free graphs, and its efficiency on faster than both tuning-based adaptation and the text processing and inference required by LLMs.

## 2 RELATED WORK

The development of GFMs can be understood through two lenses: **the core techniques** used in its architecture and **the target learning paradigm**, which defines how a model adapts to new tasks. This section reviews the field along these lines, showing how the limitations of current techniques motivate a paradigm shift towards true in-context learning.

### 2.1 POPULAR TECHNIQUES IN GFM ARCHITECTURE

To address key challenges like graph heterogeneity and task adaptation, researchers have developed several powerful techniques that are often used in combination.

**LLMs for GFMs** A primary research direction for building GFMs involves unifying heterogeneous graph data within the text domain to leverage the capabilities of LLMs. These efforts can be broadly grouped into two strategies. The first uses an **LLM as an enhancer**, processing diverse textual node features into a common representation space before performing structure-aware prediction (Huang et al., 2023; Liu et al., 2024; Chen et al., 2023; Li et al., 2024b; He et al., 2024; Wang et al., 2024c; Plenz & Frank, 2024; Zhu et al., 2025b; Xia et al., 2024). For instance, ZeroG (Li et al., 2024b) leverages an LLM to encode textual node attributes into a unified semantic space and then performs neighborhood aggregation for the final prediction. The second strategy employs the **LLM**

Table 1: Comparison of GILT with representative Graph Foundational Model paradigms.

| Method / Paradigm | LLM-Free | Tuning-Free | Multi-Domain Pre-training | Multi-Task | Few-shot |
|---|---|---|---|---|---|
| ZeroG | ✗ | ✓ | ✓ | ✗ | ✗ |
| GOFA | ✗ | ✓ | ✓ | ✓ | ✗ |
| GCOPE | ✓ | ✗ | ✓ | ✓ | ✓ |
| RiemannGFM | ✓ | ✗ | ✓ | ✓ | ✓ |
| OFA | ✗ | ✓ | ✓ | ✓ | ✓ |
| GraphAny | ✓ | ✓ | ✗ | ✗ | ✓ |
| **GILT (Ours)** | ✓ | ✓ | ✓ | ✓ | ✓ |

**as the predictor**, aiming to leverage its generative capabilities for greater task flexibility (Tang et al., 2024; Chen et al., 2024; He et al., 2025a; Zhang et al., 2024b; Kong et al., 2025; Hu et al., 2024; He et al., 2025b; Wang et al., 2024a;b; Zhang et al., 2024a; Sun et al., 2025b). The GOFA model (Kong et al., 2025) achieves this by interleaving GNN layers into an LLM's architecture, combining message-passing with semantic reasoning before generating prediction. Ultimately, the reliance of both strategies on a textual foundation naturally limits their scope to text-attributed graphs.

**Graph Prompting** Another dominant technique is graph prompting, which leverages the universal importance of topology by introducing learnable structural components that steer a pre-trained model's reasoning across diverse tasks. This paradigm has largely shifted from costly full fine-tuning towards more parameter-efficient graph prompting for task adaptation. Following an initial large-scale pre-training phase, the GNN's weights are frozen. Adaptation is then achieved by introducing and tuning small, learnable prompts that steer the model's behavior for new tasks (Sun et al., 2022; Liu et al., 2023; Sun et al., 2023a; Fang et al., 2023; Zi et al., 2024; Zhao et al., 2024a; Sun et al., 2025a; Zhao et al., 2024b; Yu et al., 2025a; Zhu et al., 2025a; Lin et al., 2025; Yu et al., 2025b). For instance, the GCOPE (Zhao et al., 2024a) framework extends a prompt-like mechanism to the pre-training stage, using learnable coordinators as virtual nodes to align different graph datasets. The more recent RiemannGFM (Sun et al., 2025a) introduces a novel geometric perspective, pre-training a model on a universal "structural vocabulary" of trees and cycles, which is then adapted to new tasks through prompt tuning. While parameter-efficient, these methods' adaptation process still hinges on **gradient-based updates for each new graph**.

## 2.2 THE PARADIGM SHIFT TOWARDS IN-CONTEXT LEARNING

The ultimate goal for a GFM is to operate as a ready-to-use system that generalizes to new tasks without re-training. This has motivated a shift towards In-Context Learning, a paradigm where a pre-trained model solves a new task at inference time using only a few prompted examples, without any parameter updates (Brown et al., 2020). While popularized by LLMs, the success of ICL on structured tabular data (Hollmann et al., 2022; 2025), computer vision (Wang et al., 2023), and time-series (Lu et al., 2025) has underscored its potential for more modalities.

Pioneering frameworks have begun to explore this direction. OFA (Liu et al., 2024), for instance, enables in-context learning through constructing a unified prompt graph connecting labeled support examples to their class nodes, allowing a GNN to synthesize this structural context for classification in a single forward pass. More recently, GraphAny (Zhao et al., 2025) achieved tuning-free generalization by using a pre-trained attention module to fuse the outputs of multiple non-parametric, analytical solvers. These works established the viability of tuning-free adaptation on graphs and laid the groundwork for more advanced ICL systems.

Building on these insights, GILT introduces a more general and powerful framework for in-context learning on graphs which uses a Transformer backbone chosen for its proven strength in ICL across language and tabular domains. With the aid of a specialized graph encoding module, the framework is inherently LLM-free to learn from raw numerical features alone. Its deep, pre-trained model is designed to learn complex non-linear patterns, making it a flexible solution for node, link, and graph classification tasks.

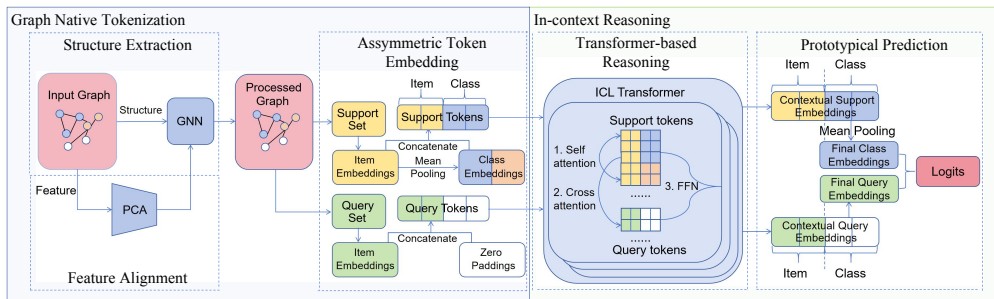

Figure 1: GILT begins with a **Graph-Native Tokenization** module converting a few-shot task into unified tokens. This module first aligns feature dimensions, then uses a GNN to generate structure-aware embeddings. These embeddings are then combined with class prototypes to form the support and query tokens. The tokens are then passed to ICL Transformer, which features a two-stage attention mechanism for in-context reasoning and a Prototypical Head for the final classification.

## 3 METHOD

To overcome the text and tuning barriers inherent in current GFMs, we introduce the GILT framework. The core technical contribution of our work is to reframe the few-shot graph learning task as a problem of reasoning over a set of contextual tokens, allowing us to leverage the power of the Transformer for universal in-context reasoning on graphs.

We formally define a graph as $G = (V, E)$ with node features $X \in \mathbb{R}^{|V| \times d_{in}}$ and an adjacency matrix $A$. GILT is designed for the **few-shot, in-context learning** setting. Each task is an $N$-way $K$-shot problem, where the model is given a **support set** of labeled examples, $\mathcal{S} = \{(x_i, y_i)\}_{i=1}^{N \times K}$, and must predict labels for an unseen **query set**, $\mathcal{Q} = \{x_j\}_{j=1}^{Q}$. The items $x_i$ can be nodes, edges, or entire graphs, depending on the task. The model must leverage the context from $\mathcal{S}$ to make predictions for $\mathcal{Q}$ without any parameter updates.

### 3.1 ARCHITECTURE OVERVIEW

As illustrated in Figure 1, the GILT framework is a two-phase pipeline designed to first translate any graph task into a universal format and then reason over that format to make predictions.

**Phase 1: Graph-Native Tokenization (Syntactic Unification).** The first phase tackles graph heterogeneity. It converts a raw, few-shot graph task with its unique features and structure into a standardized set of contextual tokens. This creates a unified format that represents the problem.

**Phase 2: In-Context Reasoning (Semantic Unification).** The second phase is designed to understand the task's meaning from the tokens alone without any tuning. A specialized ICL Transformer processes the token set, learning the task's rules from the provided examples to make a prediction.

### 3.2 GRAPH-NATIVE TOKENIZATION

The Graph-Native Tokenization phase performs **syntactic unification**, converts raw graphs and the few-shot task definition into the set-based representation required by the ICL reasoning phase. This section details the components of this phase: the alignment of feature dimensions, the extraction of structural information, and the final task-specific asymmetric token formulation.

**Alignment of Feature Dimension.** A universal GFM must accept graphs with arbitrary feature dimensions. To solve this, we first project all node features into a fixed-$d$-dimensional space. We employ a non-parametric approach using Principal Component Analysis (PCA): features with more than $d$ dimensions are reduced to $d$ principal components, while features fewer than $d$ are transformed and then zero-padded. This yields a unified feature matrix $X' \in \mathbb{R}^{|V| \times d}$ for any input graph, which undergoes a final column-wise standard scaling before being passed to our structural encoder.

**Extraction of Structure Information.** This scaled feature matrix is then enriched with topological information by our structural encoder, a deep, multi-layer Graph Convolutional Network (GCN). Critically, we employ a **linear version of the GCN**, omitting the learnable weight matrices and non-linear activations. This design aligns with simplified GCN architectures such as SGC (Wu et al.,

2019) and APPNP (Klicpera et al., 2019), which demonstrate that stripping away non-linearities can mitigate overfitting while effectively capturing structural information. The rationale for this choice is rooted in our design principle: a learnable projection at this stage tends to overfit the feature semantics of the pre-training graphs, hindering generalization. By using a simple, parameter-free aggregator, we ensure its role is to strictly extract local structural patterns, deferring all complex semantic reasoning to the more powerful ICL Transformer. We also find that a deeper encoder (4-6 layers) provides a richer, multi-hop context that is highly beneficial for the downstream Transformer. To stabilize the representations as they propagate through the deep encoder, each linear aggregation is followed by an independent LayerNorm with its own learnable affine parameters:

$$H^{(l+1)} = \text{LayerNorm}(\tilde{A}H^{(l)}) \tag{1}$$

where $\tilde{A}$ is the normalized adjacency matrix with self-loops and $H^{(0)} = X'$. The output of the final layer is the node embedding matrix $H \in \mathbb{R}^{|V| \times d}$.

**Asymmetric Token Formulation.** The final and most crucial step is to encode the entire N-way K-shot task into a set of fixed-dimension tokens. We derive a single vector representation, denoted $h$, for each task item (a node's embedding for node tasks, an element-wise product for link tasks, or a pooled vector for graph tasks). These item representations, $h_i$, are then used to form the final tokens through an asymmetric process. An initial representation for each class $p_c$ is computed using simple **mean pooling** over the support item representations, followed by L2 normalization for stability. This class representation is then paired with each support item's representation to form the final support tokens, while query item representations are paired with zero-padding:

$$\mathbf{p}_c = \frac{\frac{1}{|\mathcal{S}c|} \sum_{(x_i, y_i) \in \mathcal{S}_c} \mathbf{h}_i}{\left\| \frac{1}{|\mathcal{S}c|} \sum_{(x_i, y_i) \in \mathcal{S}_c} \mathbf{h}_i \right\|_2}, \quad \mathbf{t}_s = [\mathbf{h}_i || \mathbf{p}_{y_i}], \quad \mathbf{t}_q = [\mathbf{h}_j || \mathbf{0}] \tag{2}$$

This prototypical formulation solves a core challenge. Alternatives like one-hot encoding would result in a *variable token dimension*, while decomposing the task into binary problems would prevent the model from reasoning about *inter-class relationships*. Our approach ensures a consistent token size while enabling the Transformer to reason over all class concepts in a shared context.

### 3.3 IN-CONTEXT REASONING

With the task syntactically unified into a set of tokens, the In-Context Prediction phase is designed to perform the more complex challenge of **semantic unification**. This is accomplished by two key components: The ICL Transformer performs the core contextual reasoning, learning a task-specific mapping from the prompted examples. This is followed by the Prototypical Head, which provides a dynamic and tuning-free classification mechanism adapting to any N-way task.

**Transformer-based Reasoning.** The ICL Transformer consists of a stack of $L$ identical layers designed to process the unified set of support and query tokens and produce context-aware embeddings. The design is inspired by the principle of causal attention masking, which has been proven essential for in-context learning on structured tabular data by TabPFN (Hollmann et al., 2025). This principle ensures the query items do not influence the representation of the support set, nor should they influence each other. We implement this via a specialized **two-stage process**.

**Stage 1: Context Refinement:** The first stage of each layer builds a rich, task-specific context from the support set. To achieve this, a multi-head self-attention mechanism is applied exclusively to the set of support tokens, $T_\mathcal{S}$. This allows the support examples to interact and form a coherent representation of the task's semantics. The output is a set of refined support embeddings, $T'_\mathcal{S}$:

$$T'_\mathcal{S} = \text{SelfAttention}(T_\mathcal{S})$$

**Stage 2: Information Gathering:** This stage uses the refined context to inform the representation of the query tokens. This is the core in-context learning step where the model applies its understanding to the prediction targets. We use a multi-head cross-attention mechanism where the query tokens $T_\mathcal{Q}$ serve as the query, while the refined support embeddings $T'_\mathcal{S}$ serve as both the keys and values:

$$T'_\mathcal{Q} = \text{CrossAttention}(Q = T_\mathcal{Q}, K = T'_\mathcal{S}, V = T'_\mathcal{S})$$

A key architectural choice in our design is that both of these attention operations are performed by **a shared attention module**. Similarly, after the attention stages, all resulting token embeddings are processed by **a single, shared Feed-Forward Network (FFN).** This extensive weight sharing within each layer forces the model to learn a unified and general-purpose reasoning process. Our experiments confirm this design is highly effective for generalization. As is standard, residual connections and LayerNorm are applied around each sub-module to ensure stable training.

**Prototypical Prediction.** The final stage of our framework is the Prototypical Head, which performs the final, tuning-free classification. Our architecture is designed to maintain a separation of roles within each token embedding: the "item space" serves as the primary input for reasoning, while the "class space" serves as the dedicated output space. The ICL Transformer learns to project its final, context-aware prediction into this class-space portion of its output embeddings. Therefore, for the final prediction, **we use only the class-space portion of the embeddings from the Transformer**. The final, tuning-free classification then proceeds as follows: First, a definitive prototype vector $p_c$ for each class $c$ is computed by taking the element-wise mean of the class-space portions of all final support embeddings. Each query's final class-space embedding is then classified based on its cosine similarity to each class prototype, and these scores are converted into a probability distribution via a softmax function. This entire mechanism is non-parametric and allows GILT to adapt to any N-way classification task on the fly.

### 3.4 Pre-training

GILT is not trained to solve specific tasks but is instead taught the general meta-skill of in-context learning with **only one unified model**. The goal of pre-training is to optimize its parameters to become an effective few-shot reasoner. To achieve this, we use a diverse collection of 15 datasets from domains like citation, social, and molecular networks. This pre-training data totals over 450,000 nodes and 4 million edges, with individual graph sizes ranging from tens to over 170,000 nodes. GILT learns via a multi-task objective covering node, link, and graph classification, on features with dimensions varying from single digits to over 8,000. Further details are in Appendix B.1.

At each training step, a complete $N$-way $K$-shot few-shot task consisting of a support set $\mathcal{S}$ and a query set $\mathcal{Q}$ is generated from our diverse training corpus. This task is then formatted and passed through the GILT architecture to produce predictions. A standard cross-entropy loss is computed between these predictions and the ground-truth labels of the query items, $y_j$:

$$\mathcal{L} = -\frac{1}{|\mathcal{Q}|} \sum_{x_j \in \mathcal{Q}} \log(P(y = y_j | x_j)) \tag{3}$$

This loss is then backpropagated to update all learnable parameters in the model. By optimizing the model over millions of such task instances, the framework is not trained to memorize specific graphs or labels. Instead, it is explicitly trained to learn the meta-skill of **inferring a task's rules from a given support set and applying them to a query set**, thereby acquiring its ability to perform in-context generalization on completely unseen graphs.

## 4 Experiment

We conduct comprehensive experiments to evaluate GILT across three fundamental graph learning tasks: node classification, link prediction, and graph classification. The core objective is to assess its few-shot performance on unseen graphs against a suite of contemporary GFMs.

### 4.1 Experiment Setup

**Datasets.** To ensure a fair comparison, we selected datasets that are canonical for their respective tasks in the literature. For **node classification**, we use the widely-cited `Cora`, `Citeseer`, `Pubmed` (Yang et al., 2016), and `WikiCS` (Mernyei & Cangea, 2020) benchmarks. These offer broad coverage of published results across the baseline models. For **link prediction**, we again use the Planetoid datasets and add the `ogbl-collab` (Hu et al., 2020a) benchmark for large-scale evaluation. For **graph classification**, we employ standard OGB benchmarks (Hu et al., 2020a), `ogbg-molhiv` and

Table 2: **Few-shot Node Classification Performance.** We report 1-shot and 5-shot accuracy (%) compared against (a) state-of-the-art Few-Shot GFMs, and (b) Fully Supervised Reference Models.

**(a) Comparison with Few-shot GFMs.**

| Model | Cora | | Citeseer | | Pubmed | | WikiCS | | Average | |
|---|---|---|---|---|---|---|---|---|---|---|
| | 1-shot | 5-shot | 1-shot | 5-shot | 1-shot | 5-shot | 1-shot | 5-shot | 1-shot | 5-shot |
| RiemannGFM | 25.08 ± 9.52 | 46.82 ± 15.73 | 27.22 ± 9.21 | 36.52 ± 13.56 | 44.36 ± 8.48 | 57.28 ± 6.68 | 49.85 ± 3.30 | 53.21 ± 4.20 | 36.63 | 48.46 |
| GCOPE | 39.05 ± 1.73 | 67.06 ± 1.41 | **55.87 ± 0.23** | **63.90 ± 1.37** | 40.89 ± 1.22 | 64.34 ± 1.94 | 38.72 ± 0.40 | 47.73 ± 1.29 | 43.63 | 60.76 |
| OFA | 30.52 ± 0.62 | 41.30 ± 1.89 | 40.86 ± 0.17 | 52.01 ± 1.12 | 31.07 ± 0.85 | 37.70 ± 0.66 | 38.50 ± 1.12 | 49.23 ± 0.50 | 35.24 | 45.06 |
| GraphAny | 49.30 ± 5.95 | **72.68 ± 2.47** | 42.66 ± 8.30 | 62.08 ± 4.98 | **56.48 ± 8.98** | **69.54 ± 2.75** | 51.12 ± 7.79 | 57.86 ± 9.53 | 49.89 | 65.54 |
| **GILT (Ours)** | **52.48 ± 6.38** | 70.58 ± 2.75 | 45.40 ± 5.38 | 61.44 ± 1.57 | 41.14 ± 6.50 | 64.96 ± 7.48 | **61.44 ± 1.36** | **69.40 ± 2.02** | **50.12** | **66.60** |

**(b) Comparison with Supervised Reference Standards (Train on full train split).**

| Model | Cora | Citeseer | Pubmed | WikiCS | Protocol |
|---|---|---|---|---|---|
| GCN (Supervised) | 81.40 ± 0.70 | 63.40 ± 0.63 | 76.60 ± 0.32 | 79.12 ± 0.45 | Tuned (100% Data) |
| GAT (Supervised) | 81.70 ± 1.43 | 69.10 ± 1.59 | 77.30 ± 0.60 | 79.29 ± 0.20 | Tuned (100% Data) |
| *GILT (5-shot)* | *70.58 ± 2.75* | *61.44 ± 1.57* | *64.96 ± 7.48* | *69.40 ± 2.02* | **Tuning-Free (5-shot)** |

`ogbg-molpcba`, which are popular benchmarks for this task. Crucially, the features in these standard benchmarks are **not natural language**, but high-dimensional numerical features. The statistics are shown in Appendix A.

**Baselines** We compare GILT against a comprehensive set of baselines chosen to represent the key GFM paradigms. The competitive landscape differs significantly for each of our three target tasks, so we outline our comparison strategy for each task individually:

For **Node Classification**, we evaluate against two main groups. The first group contains our direct few-shot competitors, which are evaluated under the same N-way, K-shot protocol. This includes **Tuning-Based Models** like GCOPE (Zhao et al., 2024a) and RiemannGFM (Sun et al., 2025a), and **ICL Models** like OFA (Liu et al., 2024) and GraphAny (Zhao et al., 2025). The second group provides broader insights by comparing GILT to a range of LLM-based zero-shot baselines, including GraphGPT (Tang et al., 2024), LLaga (Chen et al., 2024), ZeroG Li et al. (2024b), UniGraph (He et al., 2025a), and GOFA (Kong et al., 2025), which leverage textual class descriptions that are unavailable in our setting. Crucially, we introduce a third group of Supervised Reference Standards, fully supervised GCN and GAT, to establish the ideal performance for each dataset. For **Link Prediction**, established few-shot GFM baselines are less common and face problems on diversed settings with different focus. We therefore mainly compare GILT against standard, fully supervised models (GCN (Kipf & Welling, 2016), GraphSAGE (Hamilton et al., 2017)) and specialized subgraph methods (SEAL (Zhang & Chen, 2018), MaskGAE (Li et al., 2023)) trained end-to-end to provide a strong performance anchor, and also include results from the recent TEA-GLM (Wang et al., 2024a) framework, which provides comparison to GFMs. For **Graph Classification**, we compare against available few-shot GFM baselines, including OFA (Liu et al., 2024) and GFT (Wang et al., 2024c). Details for baseline evaluation are shown in Appendix B.2.

**Evaluation Protocol** For all few-shot experiments, we adhere to a strict evaluation protocol to prevent data leakage. The support set is always sampled from the training split, and the query set for final evaluation is sampled from the test split. We report `Accuracy` for node classification, `Hits@K` for link prediction comparison with standard supervised GNNs (Hits@100 for Planetoid, Hits@50 for ogbl-collab) and `ROC-AUC` for left link prediction and graph classification tasks. All evaluations use the official public data splits where available. Crucially, while standard GNN baselines are trained in a fully supervised manner using all training split labels/edges, GILT is evaluated in strict 5-shot without gradient updates. Thus, the supervised baselines serve as performance upper bounds rather than direct few-shot competitors.

## 4.2 PERFORMANCE AND EFFICIENCY ANALYSIS

In this section, we present the main results, focusing on GILT's two primary strengths: its state-of-the-art few-shot performance across multiple tasks and its superior inference efficiency.

**Node Classification.** As presented in Table 2, GILT establishes itself as the state-of-the-art few-shot learner for node classification, achieving the highest average accuracy in both 1-shot (50.12%) and

Table 3: Few-shot performance on link prediction under standard and cross-task settings.

(a) Few-Shot Performance vs. Fully Supervised Baselines(Train on full train split). (Hits@K %)

| Model | Cora | Citeseer | Pubmed | ogbl-collab |
|---|---|---|---|---|
| GCN (Supervised) | $66.79 \pm 1.65$ | $67.08 \pm 2.94$ | $53.02 \pm 1.39$ | $44.75 \pm 1.07$ |
| SAGE (Supervised) | $55.02 \pm 4.03$ | $57.01 \pm 3.74$ | $39.66 \pm 0.72$ | $48.10 \pm 0.81$ |
| SEAL (Supervised) | $81.71 \pm 1.30$ | $83.89 \pm 2.15$ | $75.54 \pm 1.32$ | $64.74 \pm 0.43$ |
| MaskGAE (Supervised) | $82.48 \pm 0.76$ | $86.11 \pm 2.26$ | $\mathbf{80.38 \pm 0.82}$ | $\mathbf{65.84 \pm 0.47}$ |
| **GILT (5-shot)** | $\mathbf{85.27 \pm 1.48}$ | $\mathbf{88.53 \pm 0.72}$ | $67.26 \pm 5.47$ | $51.42 \pm 1.60$ |

(b) Few-Shot GILT vs. LLM-based Zero-Shot Baselines (AUC) under Cross Task Setting.

| Model | Pubmed | Cora | Computer | Photo |
|---|---|---|---|---|
| OFA | 0.481 | 0.492 | 0.461 | 0.459 |
| GraphGPT | 0.501 | 0.520 | - | - |
| LLaGA | 0.569 | 0.537 | 0.479 | 0.478 |
| TEA-GLM | 0.689 | 0.586 | 0.554 | 0.545 |
| GILT*(5-shot) | $\mathbf{0.867 \pm 0.014}$ | $\mathbf{0.896 \pm 0.023}$ | $\mathbf{0.854 \pm 0.033}$ | $\mathbf{0.874 \pm 0.038}$ |

* GILT is only pretrained on node-level tasks under this experiment.

5-shot (66.60%) settings. Its strength is particularly evident in challenging low-data scenarios: on WikiCS 1-shot, GILT's accuracy of 61.44% significantly surpasses all competitors, highlighting its ability to generalize from minimal context. Compared to tuning-based models, GILT is highly competitive, and in most cases superior without requiring any test-time gradient updates. Furthermore, while GraphAny is a specialized solver for node-level task only, GILT achieves a comparable elite standing as a generalist model, validating that its powerful ICL architecture is as effective as a task-specific method. Comparison with Supervised Reference Standards. While Table 2(a) demonstrates GILT's superiority over few-shot baselines, we also benchmark against fully supervised GNNs in Table 2(b) to establish a performance skyline. As expected, these reference standards achieve higher absolute accuracy. However, GILT remarkably recovers a substantial fraction of this supervised performance while operating in a strict 5-shot, zero-tuning manner, This confirms that GILT provides a viable, high-efficiency solution for scenarios where the extensive data and compute required for training a full GCN are unavailable.

**Link Prediction.** GILT's performance on link prediction significantly outperforming standard fully supervised GNNs that are trained from scratch (Table 3a). This demonstrates that its multi-domain pre-training captures highly effective and generalizable structural patterns for link formation. When compared against strong, specialist baselines like SEAL and MaskGAE, we observe a tendency driven by data scale. On larger datasets, these fully trained methods leverage the abundant supervision to establish a higher performance ceiling. However, on smaller graphs like Cora and Citeseer, GILT actually outperforms both SEAL and MaskGAE. This validates our core hypothesis: in low-data tasks where complex models may struggle to extract stable signals, GILT's massive pretraining offers a superior solution. The generalization capability is further highlighted in a challenging cross-task setting. When pre-trained exclusively on node classification tasks and evaluated few-shot on link prediction, GILT maintains a superior performance and dramatically surpasses leading LLM-based zero-shot GFMs (Table 3b). This confirms that GILT learns fundamental, transferable topological reasoning for an inherently structural task.

**Graph Classification.** The framework's strong performance extends to the difficult task of few-shot graph classification on challenging molecular benchmarks (Table 4). For instance, GILT achieves a ROC-AUC of 63.10% on ogbg-molhiv, surpassing other strong few-shot GFM baselines. This showcases GILT's ability to capture complex, holistic graph patterns from very few examples and confirms its versatility as a multi-task foundational model.

Table 4: Graph classification performance in AUC under a 5-shot setting.

| Model | ogbg-molhiv | ogbg-molpcba |
|---|---|---|
| OFA (5-shot) | $0.576 \pm 0.037$ | $0.548 \pm 0.038$ |
| GFT (5-shot) | $0.587 \pm 0.069$ | $0.593 \pm 0.068$ |
| **GILT (5-shot)** | $\mathbf{0.631 \pm 0.053}$ | $\mathbf{0.597 \pm 0.005}$ |

**Efficiency Analysis.** A critical advantage of GILT's tuning-free paradigm is its inference efficiency, which we visualize in Figure 2. The plot reveals a stark performance gap between paradigms. Tuning-Free models, including GILT and GraphAny, are clustered within the shaded sub-second performance region at the bottom of the chart. In stark contrast, Tuning-Based and Generative LLM models require tens of seconds to nearly an hour for a single task due to the overhead of gradient-based adaptation or massive model inference. This efficiency gap, spanning several orders of magnitude, underscores a crucial practical advantage of the tuning-free ICL approach and positions GILT as a more scalable solution for real-world applications.

## 4.3 ARCHITECTURAL VALIDATION AND ANALYSIS

To understand the sources of GILT's strong performance, we conduct a series of analyses to validate its key architectural components and deconstruct its in-context learning behavior.

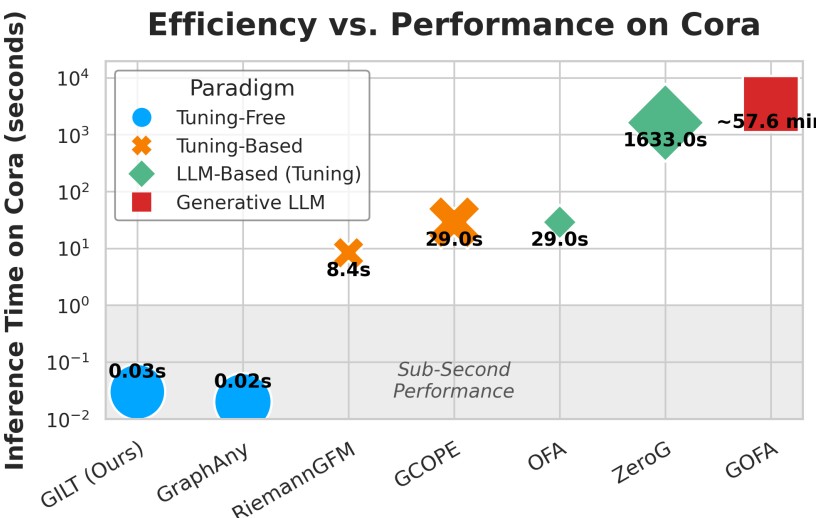

Figure 2: Efficiency vs. Accuracy on Cora node classification. The y-axis is the total inference time (lower is better) and the point size is proportional to accuracy (larger is better). All models are 5-shot, except for the LLm-based zero-shot baselines.

**What Architectural Design Makes GILT an Effective In-Context Learner?** To identify which architectural components are essential for GILT's in-context learning ability, we perform a series of ablation studies, with results presented in Table 6.

The findings first highlight the absolute criticality of the **ICL Transformer**, which acts as the core reasoning engine. Removing it entirely (*w/o ICL Transformer*) causes a catastrophic performance collapse, confirming that without this component, the model cannot perform in-context reasoning. Furthermore, deviating from our specialized design by using more conventional approaches, such as unshared attention weights or predicting with the full token, consistently degrades performance. This validates our specific choices of a shared-attention module and a dedicated class-space output for promoting generalization.

While the Transformer is the reasoning engine, the results show that generalizable structural features are a prerequisite for its success. The **Graph Encoder** is therefore equally crucial. Operating without any structural information (*w/o Graph Encoder*) results in a severe performance drop, showing that context is of little use without meaningful features. Moreover, replacing our deep, linear GCN with either a non-linear

Table 5: Comparison of GILT's 20-shot performance against LLM-based zero-shot baselines.

| Model (Setting) | Cora | Citeseer | Pubmed | WikiCS |
|---|---|---|---|---|
| GraphGPT[*] | 17.48 | 13.93 | 42.94 | 33.59 |
| LLaGA[*] | 11.62 | 19.52 | 7.56 | 10.98 |
| ZeroG[*] | 64.21 | 50.35 | 74.68 | 31.26 |
| UniGraph[*] | 69.53 | - | 72.48 | 43.45 |
| GOFA[*] | 70.81 | 65.72 | 74.76 | 71.17 |
| **GILT (20-shot)** | $78.58 \pm 1.63$ | $67.36 \pm 2.07$ | $72.46 \pm 3.05$ | $69.36 \pm 1.44$ |

[*] Results are reported from original studies, which did not include standard deviations.

or a shallow 2-layer version leads to a consistent decrease in accuracy. This highlights a key insight: while shallow, non-linear encoders are typical choices in standard supervised settings, they are suboptimal for pre-training an in-context learner. This confirms that a deep, linear architecture's advantage for extracting rich and generalizable structural patterns without overfitting to the pre-training data.

**How well does GILT learn from in-context examples?** To probe the effectiveness of the knowledge GILT acquires from context, we compare it against zero-shot LLMs. These baselines utilize explicit textual descriptions of the classes that often requires laborious pre-processing to obtain, while GILT must infer semantics from the examples alone.

Table 5 show that with a 20-shot context, GILT's performance is highly competitive with, and in several cases surpasses, strong LLM baselines. This highlights a crucial distinction in their underly-

Table 6: Ablation study results showing 5-shot accuracy (%).

| Model Variant | Cora | Citeseer | Pubmed | WikiCS |
|---|---|---|---|---|
| **GILT (Full Model)** | **70.58 ± 2.75** | **61.44 ± 1.57** | **64.96 ± 7.48** | **69.40 ± 2.02** |
| *Ablations on the ICL Transformer* | | | | |
| w/o ICL Transformer | 13.00 ± 0.00 | 7.70 ± 0.00 | 18.00 ± 0.00 | 2.51 ± 0.00 |
| w/ Unshared Attention | 67.42 ± 4.02 | 55.38 ± 4.61 | 56.28 ± 12.85 | 63.55 ± 1.31 |
| w/ Full Token for Prediction | 67.90 ± 4.77 | 55.64 ± 4.23 | 55.68 ± 11.86 | 63.68 ± 1.39 |
| *Ablations on the Graph Encoder* | | | | |
| w/o Graph Encoder | 28.72 ± 6.17 | 30.50 ± 1.69 | 42.54 ± 6.36 | 32.74 ± 4.99 |
| w/ Non-linear GCN | 65.52 ± 6.29 | 56.68 ± 4.00 | 64.60 ± 4.52 | 67.88 ± 1.98 |
| w/ 2-layer Encoder | 64.98 ± 7.28 | 57.12 ± 2.91 | 61.26 ± 6.63 | 66.24 ± 2.49 |

ing mechanisms: while LLMs perform knowledge retrieval by applying vast, pre-existing linguistic information, GILT performs fundamental, graph-native reasoning. It successfully infers a class's functional definition purely from its numerical and structural context.

## 5 CONCLUSION

We introduced GILT, a novel Graph Foundational Model designed to be both **LLM-free** and **tuning-free**. Our key innovation is reframing few-shot graph tasks as a token-based reasoning problem, allowing a pre-trained Transformer to learn from examples directly at inference time. Experiments confirm that this in-context learning approach achieves strong few-shot performance, offering a more general and efficient solution than prior methods.

A promising future direction is to advance the feature unification stage. While our current PCA-based approach is robust, a learnable dimension unifier could offer greater expressive power (Zhao et al., 2024b). Further avenues include scaling the model with larger pre-training corpora and investigating advanced objectives like contrastive learning to enhance representation quality.

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

# A  DATASET DETAILS

This section provides detailed statistics for the datasets used in our pre-training corpus and for our downstream evaluations. All evaluation datasets were strictly held out and unseen during the pre-training phase.

## A.1  PRE-TRAINING DATASETS

To learn a general-purpose in-context reasoner, GILT was pre-trained on a large and diverse corpus of 15 publicly available graph datasets. This corpus was curated to span multiple domains (social, citation, bioinformatics) and task levels (node, link, and graph) to ensure the model learns robust and generalizable patterns. A summary of the key datasets included in the pre-training corpus is provided in Table 7.

v

Table 7: Statistics of datasets used in the GILT pre-training.

| Dataset | Domain | # Graphs | # Nodes | # Edges | # Features | # Classes |
|---|---|---|---|---|---|---|
| ogbn-arxiv (Hu et al., 2020a) | Citation | 1 | 169,343 | 1,166,243 | 128 | 40 |
| CS (Shchur et al., 2018) | Co-authorship | 1 | 18,333 | 163,788 | 6,805 | 15 |
| Physics (Shchur et al., 2018) | Co-authorship | 1 | 34,493 | 495,924 | 8,415 | 5 |
| Computers (Shchur et al., 2018) | Co-purchase | 1 | 13,752 | 491,722 | 767 | 10 |
| Photo (Shchur et al., 2018) | Co-purchase | 1 | 7,650 | 238,162 | 745 | 8 |
| Flickr (Zeng et al., 2020) | Social | 1 | 89,250 | 899,756 | 500 | 7 |
| DBLP (Bojchevski & Günnemann, 2018) | Citation | 1 | 17,716 | 105,734 | 1,639 | 4 |
| USA (Ribeiro et al., 2017) | Transportation | 1 | 1,190 | 13,599 | 1,190 | 4 |
| Brazil (Ribeiro et al., 2017) | Transportation | 1 | 131 | 1,038 | 131 | 4 |
| Europe (Ribeiro et al., 2017) | Transportation | 1 | 399 | 5,995 | 399 | 4 |
| Wiki (Yang et al., 2023) | Web pages | 1 | 2,405 | 17,981 | 4,973 | 17 |
| BlogCatalog (Yang et al., 2023) | Social | 1 | 5,196 | 343,486 | 8,189 | 6 |
| FacebookPagePage (Rozemberczki et al., 2021) | Social | 1 | 22,470 | 342,004 | 128 | 4 |
| ogbg-bace (Hu et al., 2020a) | Molecular | 1,513 | 34.1 (avg) | 73.7 (avg) | 9 | 2 |
| ogbg-bbbp (Hu et al., 2020a) | Molecular | 2,039 | 23.9 (avg) | 51.6 (avg) | 9 | 2 |

## A.2 EVALUATION DATASETS

We evaluated GILT's few-shot performance on a suite of 8 unseen benchmark datasets. These datasets were chosen as they are standard in the literature and cover all three primary graph learning tasks. Detailed statistics for each evaluation dataset are summarized in Table 8.

Table 8: Statistics of datasets used in the GILT evaluation.

| Dataset | Domain | # Graphs | # Nodes | # Edges | # Features | # Classes/Tasks |
|---|---|---|---|---|---|---|
| Cora (Yang et al., 2016) | Citation | 1 | 2,708 | 10,556 | 1,433 | 7 |
| Citeseer (Yang et al., 2016) | Citation | 1 | 3,327 | 9,104 | 3,703 | 6 |
| Pubmed (Yang et al., 2016) | Citation | 1 | 19,717 | 88,648 | 500 | 3 |
| WikiCS (Yang et al., 2016) | Web pages | 1 | 11,701 | 216,123 | 300 | 10 |
| ogbl-collab (Hu et al., 2020a) | Co-authorship | 1 | 235,868 | 1,285,465 | 128 | Link Prediction |
| ogbg-molhiv (Hu et al., 2020a) | Molecular | 41,127 | 25.5 (avg) | 27.5 (avg) | 9 | 1 |
| ogbg-molpcba (Hu et al., 2020a) | Molecular | 437,929 | 26.0 (avg) | 28.1 (avg) | 9 | 128 |

## B IMPLEMENTATION DETAILS

This section provides the specific implementation details for our model, GILT, and all baselines used in the experiments.

### B.1 GILT MODEL AND PRE-TRAINING

**Architecture Details.** The GILT model evaluated in our experiments was configured with the following architecture. The non-parametric encoder projects all node features into a unified dimension of $d = 512$. The structural encoder is a 5-layer linear GCN, with each aggregation step followed by a LayerNorm with learnable affine parameters. The ICL Transformer consists of 5 layers, with 4 attention heads per layer and a hidden dimension of 4096 in the feed-forward networks.

**More Design Choices.** Several key architectural decisions were made based on preliminary experiments to enhance performance and generalization.

- Feature Alignment and Pre-processing: Our feature alignment pipeline is a multi-step process designed to robustly handle graphs with diverse feature dimensions. For graphs with high-dimensional features, we use PCA for dimensionality reduction. Conversely, to handle low-dimensional features without excessive zero-padding, we first standardize them to an intermediate dimension via PCA before applying a learnable linear projection to the final model dimension. For scalability, this PCA step is implemented using an approximate, incremental version that runs on the CPU when feature matrices are too large for GPU memory. Finally, the entire processed feature matrix undergoes a crucial column-wise standard scaling, which normalizes each feature dimension independently and prevents padded zeros from distorting the feature statistics.

- Graph Encoder: Within the linear GCN encoder, we found two important, and one counter-intuitive, results. First, including the learnable affine parameters in each LayerNorm step was essential for performance; removing them caused a significant loss. Second, we found that adding residual connections between the linear GCN layers, a standard practice in deep networks, was not effective and did not improve results.

- Prototype Formulation: For generating the class prototypes, we confirmed that L2-normalizing the vectors after mean pooling is highly beneficial for stabilizing the model. We also experimented with enforcing an additional orthogonality constraint on the prototypes but found it to be less effective than simple normalization.

**Pre-training.** The pre-training process was conducted for a total of 50 epochs, with each epoch iterating through all tasks sampled from our pre-training corpus. To improve the model's ability to learn from varying amounts of context, we employed a shot decay schedule, where the number of shots, was gradually decayed from an initial 20 down to 5 over the course of training. To enhance model robustness, we applied two forms of data augmentation: feature dropout and edge dropout. The final multi-task loss is a weighted sum of the individual task losses, with a hyperparameter controlling their relative importance. For link prediction tasks specifically, we used a negative sampling ratio of 3:1, sampling three negative links for each positive link for both the support set construction and the query set loss calculation.

## B.2 BASELINE SETUP

For our experiments, we made a distinction between re-evaluating baselines and citing established results. Specifically, for the node classification task, we conducted a fresh evaluation of all Tuning-Based and ICL baselines using their official public codebases to ensure a direct and fair comparison under our strict few-shot protocol. For all other results, including baselines on other tasks and the supervised models, performance is reported from their original publications or other well-established literature to ensure consistency with community standards.

**Tuning-Based Models.** For GCOPE (Zhao et al., 2024a), we used its official implementation and default parameters. To ensure a strict separation between pre-training and evaluation data, we modified its pre-training corpus to exclude the Planetoid datasets. During few-shot adaptation, we followed standard procedure by freezing the GNN backbone and only tuning the prompt module. For RiemannGFM (Sun et al., 2025a), while also using its default parameters, we observed that its original prediction mechanism utilizes external class information. To ensure a fair comparison focused solely on the ability to learn from the provided examples, we replaced its final head with a simple linear classifier that was then tuned on the few-shot support set.

**ICL Models.** For OFA (Liu et al., 2024) and GraphAny (Zhao et al., 2025), we utilized their official public codebases and pre-trained checkpoints to evaluate them in our few-shot setting. For OFA, we used the default model parameters and its standard Sentence Transformer for generating text embeddings; the checkpoint was pre-trained on a corpus including ogbn-arxiv. For GraphAny, we also used its default parameters and employed the official model checkpoint pre-trained on the ogbn-arxiv dataset for all node classification evaluations.

**LLM-Based Models.** The performance of the LLM-based zero-shot models was sourced from existing literature. For ZeroG, we report its performance on the Planetoid datasets as cited in its original publication (Li et al., 2024b), while its performance on WikiCS is taken from the evaluation presented in the GOFA paper (Kong et al., 2025). For GOFA, the results on Cora and WikiCS are from its original paper (Kong et al., 2025). The results for UniGraph are taken directly from its respective publication (He et al., 2025a). All reported metrics for GraphGPT and LLAGA, as well as the Citeseer and Pubmed results for GOFA, are sourced from the comprehensive benchmark evaluation in Wang et al. (2025).

**Link Prediction & Graph Classification Baselines.** For the supervised link prediction baselines, the performance of GCN (Kipf & Welling, 2017) and GraphSAGE (Hamilton et al., 2017) is reported from Chamberlain et al. (2023). For the graph classification baselines, the results for OFA (Liu et al., 2024) and GFT (Wang et al., 2024c) are sourced directly from their respective original publications.

Table 9: Hyperparameter search space and optimal values used for the GILT model.

| Hyperparameter | Search Space | Optimal Value |
|---|---|---|
| *Pre-training Hyperparameters* | | |
| Optimizer | Adam, AdamW | AdamW |
| Learning Rate | [1e-6, 1e-4] | 2e-6 |
| Weight Decay | [1e-6, 1e-2] | 4e-4 |
| Total Training Epochs | 50, 100 | 50 |
| LR Schedule | Cosine Decay, Linear Decay, WarmUp | Linear Decay |
| *Multi-Task Training Details* | | |
| Batch Size (Node Tasks) | {4096, 8192, 16384} | 8192 |
| Batch Size (Link Tasks) | {8192, 16384, 32768} | 16384 |
| Batch Size (Graph Tasks) | {1024, 2048, 4096} | 1024 |
| Loss Weight (Node) | {[0.1, 10]} | 0.53 |
| Loss Weight (Link) | {[0.1, 10} | 2.74 |
| Loss Weight (Graph) | {0.5, 1.0, 1.5} | 0.42 |
| *GILT Architecture Hyperparameters* | | |
| Unified Dimension (d) | 128, 256, 512 | 512 |
| GCN Encoder Layers | [2, 8] | 5 |
| Transformer Layers (L) | [2, 8] | 5 |
| Transformer Heads | 1, 4, 8 | 4 |
| *Regularization & Other Details* | | |
| Feature Dropout | [0, 0.5] | 0.1 |
| Edge Dropout | [0, 0.5] | 0.1 |
| Model Dropout | [0, 0.5] | 0.1 |

## B.3 COMPUTATIONAL RESOURCES

All experiments were conducted on a Linux server equipped with 8 NVIDIA RTX 4090 GPUs. Our implementation is built using PyTorch (Paszke et al., 2019) and PyTorch Geometric (PyG) (Fey & Lenssen, 2019).

## C HYPERPARAMETER SETTINGS

This section details the hyperparameter settings for GILT. To efficiently find a robust configuration, we performed a hyperparameter search using Bayesian optimization. The search was conducted on a held-out set of validation tasks sampled from our pre-training datasets to identify a single, robust set of parameters. The final optimal values, listed in Table 9, were then frozen and used for all reported experiments across all datasets and tasks.

## D INVESTIGATION INTO THE INFLUENCE OF SHOT NUMBER

To better understand how GILT utilizes contextual information, we analyzed its performance across a varying number of support examples (shots) for the node classification task. The results are presented in Figure 3.

As illustrated in the figure, there is a clear positive correlation between the number of shots provided in the context and the model's overall performance. The most significant gains in accuracy are typically observed when increasing the shot number from one to ten. As more examples are added to the context, the performance continues to improve, though the rate of improvement gradually diminishes. This analysis confirms that providing a richer context with more examples is an effective method for enhancing GILT's predictive accuracy, which is consistent with the expected behavior of an in-context learning framework.

Figure 3: The influence of the number of shots (K) on GILT's few-shot performance. The x-axis represents the number of support examples per class, and the y-axis represents the classification accuracy on the test set. Each line corresponds to a different dataset.

# E    LLM USAGE

We utilized LLMs to assist in the preparation of this work. Specifically, we used LLMs for debugging code snippets and for proofreading and improving the clarity of the manuscript's text. The authors reviewed and edited all LLM-generated content and take full responsibility for the final submission.

