# OpenReview forum: "GILT: An LLM-Free, Tuning-Free Graph Foundational Model for In-Context Learning"
_ICLR.cc/2026/Conference — Submitted to ICLR 2026_

### Official Review · Reviewer_FVan · 2025-10-28

**Soundness:** 2
**Presentation:** 2
**Contribution:** 2
**Rating:** 4
**Confidence:** 3

**Summary:**

This paper introduces GILT (Graph In-context Learning Transformer), which aims to be a tuning-free and LLM-free Graph Foundational Model (GFM). The idea is to reframe few-shot graph learning — including node, edge, and graph-level classification — as a token-based in-context reasoning problem. GILT tokenizes graph structures into a unified format, processes them using a Transformer designed for in-context learning, and predicts through a non-parametric prototypical head, avoiding gradient updates or external text encoders altogether.

**Strengths:**

1. The inference efficiency comparison (e.g., sub-second vs. minutes for tuning-based methods) is convincing. This aspect gives the work genuine practical value, especially for large-scale or latency-sensitive applications.
2. The paper is clearly written, with logical flow and professional presentation. It does a good job positioning GILT relative to prior work and articulating its contributions clearly.
3. The motivation is good — trying to build a graph foundation model that’s both LLM-free and tuning-free addresses two major bottlenecks in the field. The paper’s goal of unifying different graph tasks into a single in-context reasoning framework feels visionary and aligns well with the broader trend of foundation models in other domains.

**Weaknesses:**

1. The improvements over strong baselines like GraphAny or GCOPE are small — often within error margins. While GILT shows consistency, it doesn’t convincingly outperform other methods across all settings. The authors emphasize efficiency and generality, but in real performance terms, the margin is not enough to justify the added architectural complexity.
2. The claim that GILT performs “in-context learning” is not fully convincing. The mechanism seems closer to a meta-learning framework (learn to classify from support sets) than true in-context reasoning as understood in LLM literature. There’s no evidence that GILT dynamically infers rules beyond standard metric-based few-shot learning. This makes the “Transformer ICL” narrative feel more like rebranding than substance.
3. Using PCA for feature alignment is elegant but shallow. It’s unclear whether this linear preprocessing truly generalizes across heterogeneous graphs or just works for the small set of benchmarks. A non-learnable feature transformation might actually cap performance — and the authors themselves acknowledge this as a limitation in the conclusion.

**Questions:**

1. How is “in-context learning” here fundamentally different from conventional few-shot meta-learning?
2. Is PCA alignment fixed across datasets, or recalculated per graph? If the latter, does that leak test information?

---

> ### Author Response · Authors · 2025-12-03
>
> [W1] We appreciate the reviewer’s assessment of the performance margins. We acknowledge that on certain datasets, GILT’s improvement over strong baselines like GraphAny or GCOPE is incremental rather than transformative. However, we respectfully argue that evaluating GILT solely on absolute accuracy overlooks the the multi-dimensional challenge of the Foundation Model landscape, where the goal is to optimize the Generality-Efficiency-Performance Frontier.
>
> The reviewer suggests the performance does not justify the added architectural complexity; we argue that GILT actually achieves a massive reduction in operational complexity. Baselines like GCOPE, while architecturally standard, impose a heavy operational burden because every new task requires establishing a training loop and performing gradient updates. In contrast, GILT functions as a pure inference engine. Furthermore, we highlight the achievement of multi-level generalization. Typically, generalist models pay a performance penalty compared to specialist models due to negative transfer. GraphAny is highly specialized for Node Classification, and GCOPE utilizes task-specific tuning to maximize performance. GILT matches or exceeds these specialized baselines without their specific limitations. This robustness across diverse tasks (Node, Link, and Graph) using a single frozen set of weights is a capability that neither the node-centric GraphAny nor the tuning-dependent GCOPE possesses, justifying the architectural design.
>
> [W2 & Q1] We appreciate the reviewer’s query regarding the distinction between our ICL framework and conventional meta-learning. We clarify that our terminology aligns with the definition established in the foundation model literature[1], where ICL is defined as adaptation via the forward pass without parameter updates. "Conventional" few-shot meta-learning (most notably MAML and its variants) relies on optimization-based adaptation: the model processes the support set to calculate gradients and update its weights. GILT freezes all parameters and relies on the attention mechanism to dynamically aggregate task information from the context. The reviewer suggests the mechanism is similar to "metric-based few-shot learning." However, in standard metric learning, the rule for classification is static: a class is represented by the mean of its support samples. GILT moves beyond this by using the Transformer to dynamically infer the class structure from the context. Specifically, the Context Refinement stage allows the model to assess the coherence of the support set. By attending to the consensus of the group, the model effectively learns to down-weight outliers and amplify consistent structural patterns within that specific episode. This means the Class Prototype in GILT is not a static average, but a contextually computed representation that adapts to the noise profile of the provided examples.
>
> [W3] Thank you for your insight. We agree that the PCA transformation is linear and non-learnable. However, we demonstrate that in the cross-domain few-shot task, this shallowness is a design feature that ensures robustness. As detailed in our General Response, sophisticated alignment methods (such as FUG) frequently suffer from negative transfer and training instability. We refer the reviewer to Table in the General Response, which empirically demonstrates that our fixed PCA approach outperforms the learnable FUG framework. This confirms that while a non-learnable transformation theoretically caps performance, it provides a much higher empirical floor by avoiding the optimization instability that plagues learnable adapters in low-data settings.
>
> [Q2] Thanks for this sharp question. We clarify that PCA is indeed recalculated for the target domain, but that follows the standard transductive or inductive protocol for graph learning and does not constitute data leakage. For node classification tasks, the premise of processing query nodes in isolation is incompatible with the fundamental mechanics of GNNs, which require the the features of neighbors to perform message passing. Consequently, all standard baselines in both transductive (GCN, SAGE) and inductive settings (GCOPE, RiemannGFM) invariably utilize the full feature matrix. GILT follows this established protocol. Since this calculation relies exclusively on feature and never accesses the ground-truth labels, it functions as a standard form of unsupervised domain adaptation. Similarly, for graph classification, calculating PCA statistics on the support set alone would result in a unstable, rank-deficient covariance estimation that collapses the feature space. Therefore, it is necessary to use the unlabeled test batch to ensure a stable coordinate system. This aligns with standard domain adaptation practices where the model is permitted to access the marginal feature distribution of the target domain.
>
> [1]: Brown, Tom, et al. "Language models are few-shot learners." NeurIPS 2020

---

### Official Review · Reviewer_uE5v · 2025-10-28

**Soundness:** 2
**Presentation:** 3
**Contribution:** 3
**Rating:** 2
**Confidence:** 3

**Summary:**

The paper proposed GILT, a GFM without LLM or tuning for few-shot ICL on graph tasks. They proposed a tokenization model (PCA+GCN) that aligns different feature spaces and encode to a unified space. A ICL transformer is then used to perform token-based reasoning with causal attention and a prototypical prediction head. The authors pretrained this model with many datasets & tasks, and evaluated on different tasks. The results showed good performance as well as efficiency of the proposed method.

**Strengths:**

1. I think is paper showed a interesting new direction of aligning different graphs without pretrained LLMs, which loosened the constrain of text-attributed graphs of many of GFM works, while still not requiring per-graph tuning for the unseen feature spaces.
2. The paper is overall clearly written and easy to follow.
3. The experimental design is thorough, with good focus on practicalness (efficiency).

**Weaknesses:**

1. While PCA is a handy tool to convert arbitrary feature space into a fixed dimension, I think it's unconvincing to rely on it for feature alignment. PCA's only goal is to preserve variance, and it would create a new feature space for each of the graphs (although they are all $d$-dim), so the feature spaces are still independent. Additionally, PCA would probably throw away lots of useful information that are task relevant, given how it works. I'm surprised on how well this method works with PCA being the foundational feature alignment component. And this also makes me wonder how much does node features actually contribute in this system - what if we just swap all features in all graphs with random $d$-dim vectors?

**Questions:**

1. The authors highlighted multiple times that the tokenizer is tuning-free for new graphs. But in Appendix B.1 (line 889), the authors mentioned that there is a learnable linear projector for some of the graphs.

---

> ### Author Response · Authors · 2025-12-03
>
> We thank the reviewer for the sharp intuition. We address the three interconnected concerns below.
>
> (1) On Semantic Independence: We acknowledge that PCA creates semantically independent feature spaces for each graph. However, as discussed in the General Response, our results demonstrate that strict semantic alignment is not required for effective few-shot transfer. We refer the reviewer to the General Response for the detailed discussion on Spectral Denoising and the comparison against learnable baselines.
>
> (2) On "Throwing Away" Information: We appreciate the reviewer's theoretical concern, but we argue that practically many standard graph feature sets are inherently highly redundant. PCA functions as an effective compression mechanism that retains the vast majority of signal energy in the top-k components. To quantify this, we conducted a controlled ablation using a standard semi-supervised GCN trained on both Raw Features and PCA-reduced features. The results reveal that the performance gap is statistically minimal. This empirically demonstrates that the vast majority of discriminative signal is concentrated within the top-100 principal components.
>
> | Dataset | No PCA | PCA (100-dim) | Difference |
> | :--- | :--- | :--- | :--- |
> | Cora | $81.13 \pm 0.48$ | $79.60 \pm 0.78$ | $-1.53$ |
> | Citeseer | $68.13 \pm 0.74$ | $67.77 \pm 0.79$ | $-0.37$ |
> | Pubmed | $76.93 \pm 0.61$ | $76.73 \pm 0.05$ | $-0.20$ |
>
> (3) On Random Features: To answer the reviewer's specific question, we conducted this exact experiment. As shown in Table R1 of the General Response, replacing node features with random Gaussian vectors causes the model performance to collapse to near-random guessing. This empirically proves that the node features contribute crucial information.

---

### Official Review · Reviewer_h6fb · 2025-10-31

**Soundness:** 2
**Presentation:** 2
**Contribution:** 2
**Rating:** 4
**Confidence:** 4

**Summary:**

This paper proposes GILT, an LLM‑free and tuning‑free graph foundational model for few‑shot adaptation via in‑context learning. The method first converts any N‑way K‑shot graph task into a unified token set. Then, a two‑stage Transformer performs self‑attention over support tokens and cross‑attention from queries to the refined support, with a prototypical head operating in a designated class‑space for cosine‑similarity classification. On several graph benchmarks, GILT reports SOTA few‑shot node accuracy, strong link prediction and competitive graph classification, while achieving sub‑second inference compared to tuning‑based and LLM baselines.

**Strengths:**

S1: This paper focuses on an important research problem, addressing both LLM‑free and tuning‑free generalization in Graph FMs.

S2: This paper proposes a clean tokenization, Transformer ICL, and prototypical head pipeline, with a principled causal attention schedule that mirrors best practices in tabular ICL.

S3: This paper reports strong few‑shot results across tasks, including SOTA averages on 1‑shot/5‑shot node classification and competitive link/graph results.

**Weaknesses:**

W1: This paper's novelty appears incremental relative to well-known prototypical few-shot learning and set-based attention architectures; the work would benefit from deeper comparisons to similar works, such as Prototypical Networks[1] and Set Transformer[2] families.

W2: This paper standardizes feature dimensions with PCA and zero-padding before the structural encoder, but it does not justify why PCA is preferable to other linear bases (e.g., truncated SVD) or to learnable, structure-aware adapters (e.g., GNNs) that could better preserve discriminative signals under heterogeneity. A comparison or ablation (PCA vs. SVD vs. a learnable GNN) would clarify whether PCA contributes uniquely to the reported gains.

W3: This paper's "deep linear GCN" removes learned weights and nonlinearities to avoid overfitting, which is similar to SGC or APPNP. The paper should provide appropriate citations and discussions.

W4: This paper applies LayerNorm with learnable affine parameters after each linear aggregation, which is not the prevailing normalization choice in message passing. The paper should justify this design relative to alternatives such as BatchNorm or GraphNorm[3].

W5: This paper presents an attractive pipeline, but many steps are described only at a conceptual level, leaving several crucial mechanisms (e.g., attention masks, token construction, and prototype computation) insufficiently specified.

W6: This paper's link-prediction baselines omit standard strong methods (e.g., SEAL[4] and MaskGAE[5]), so the claim of superiority over fully supervised GNNs is not yet convincing.

W7: This paper evaluates node ICL on the Planetoid family and WikiCS, which are classic but small-scale graphs with known limitations and may not be ideal benchmark choices; evaluation on larger or more diverse node-level datasets would be more convincing.

W8: This paper describes the encoder as "parameter-free" yet LayerNorm contains learnable affine parameters and the Transformer itself is learned.


Minor points:

+ The results in Table 4 seem unreasonable, as GILT is reported as 63.10 while baselines are reported as decimals around 0.58. The paper should clarify the reason behind this.
+ This paper's feature-alignment procedure uses per-graph PCA but does not specify whether statistics are computed on the full graph (risking transductive leakage) or only on train/support data, which is essential in strict inductive settings.

+ This paper's cross-task link prediction evaluation appears to reuse the same graphs (Computers, Photo) that are included in the pre-training corpus, which undermines the claim of generalization to unseen graphs and conflates task transfer with graph reuse.
+ There appears to be a typo in Equation (2), where the class support set $S_c$ is multiplied by a vector $h$.



Reference:

[1] Snell, J., Swersky, K., & Zemel, R. Prototypical Networks for Few‑Shot Learning. NeurIPS 2017.

[2] Lee, J., Lee, Y., Kim, J., Kosiorek, A., Choi, S., & Teh, Y. W. Set Transformer: A Framework for Attention‑based Permutation‑Invariant Neural Networks. ICML 2019.

[3] Cai, T., et al. GraphNorm: A Principled Approach to Accelerating Graph Neural Network Training. ICML 2021

[4] Zhang, M., & Chen, Y. Link Prediction Based on Graph Neural Networks. NeurIPS 2018.

[5] Li, J., et al. What's Behind the Mask: Understanding Masked Graph Modeling for Graph Autoencoders. KDD 2023.

**Questions:**

See above.

---

> ### Author Response · Authors · 2025-12-03
>
> [W1] We thank the reviewer for this comparison. While GILT shares high-level concepts with these families, we argue that our framework represents a fundamental architectural evolution designed to solve graph-specific challenges.
>
> Standard Prototypical Networks rely on a static metric learning assumption: support samples are simply averaged to form class centroids. GILT replaces this static aggregation with the ICL Transformer. Unlike Prototypical Networks, GILT allows the support tokens to interact via self-attention and allows the query to attend to this refined context before any prototype is computed. This enables the model to perform non-linear metric adaptation.
>
> While we acknowledge that Set Transformers can perform set-to-set transformations using their attention blocks, we clarify that GILT represents a fundamentally distinct architectural paradigm. The fundamental inductive bias of a Set Transformer is permutation equivariance across the entire input; every element interacts symmetrically with every other element. In contrast, GILT implements a specialized Asymmetric Reasoning Schedule designed specifically for few-shot inference. The Support tokens first interact internally and then the Query tokens attend to this stabilized context. Moreover, Set Transformers operate on sets of independent feature vectors and lack the mechanism to handle Heterogeneous Graph Topology while the Graph-Native Tokenization pipeline in GILT solves this problem. By integrating PCA alignment and structural encoding, GILT converts raw topological data into the compatible token format that the Transformer requires, bridging a gap that generic set architectures do not address.
>
> We refer the reviewer to [W2] in our reponse to reviewer B818 for a broader discussion on other similar models like OFA and TabPFN.
>
> [W2] We thank the reviewer for this thoughtful suggestion. We clarify our design choices below:
> 1. PCA vs. Truncated SVD: We clarify that PCA is mathematically equivalent to Truncated SVD applied to centered data. We prioritized PCA (centering + SVD) over raw SVD because centering the data is crucial for distance-based metric learning.
> 2. Feasibility of Learnable GNNs: We respectfully point out that a standard learnable adapter is architecturally infeasible in the cross-domain setting because the input dimension $d_{in}$ varies arbitrarily across datasets.
> 3. Comparison with FUG (Learnable Alignment): To address the reviewer's interest in learnable alignment, we compared PCA against FUG, a SOTA method that does attempt to learn a structure-aware basis transformation. As detailed in our General Response, the fixed PCA approach significantly outperforms the learnable FUG framework in few-shot transfer.
>
> [W3] We thank the reviewer for identifying the connection. We add proper citations in the revised manuscript.
>
> [W4] We thank the reviewer for this question. While we acknowledge that GraphNorm (GN) and BatchNorm (BN) are popular in supervised GNNs, they are unsuitable for our tuning-free, cross-domain setting. Standard BN relies on running statistics accumulated during training to normalize test data. In cross-domain transfer, this causes immediate covariate shift and model collapse. LayerNorm (LN) avoids this by computing statistics independently for each sample. GN relies on a learnable shift parameter $\alpha$ to control the re-injection of the global graph mean. When transferred to a heterogeneous target domain, this fixed $\alpha$ leads to negative transfer.  In contrast, while LN also contains learnable affine parameters, they serve strictly to rescale features for activation stability. Finally, it is generally not recommended to mix different normalization schemes within a single pipeline. Since our downstream reasoning module is a Transformer, employing LN in the GCN encoder ensures distributional consistency across the architecture.
>
> [W5] We thank the reviewer for highlighting areas where the description of our pipeline could be clearer. To clarify, the Token Construction mechanism is explicitly asymmetric: as defined in Equation 2, support tokens are formed by concatenating the item embedding $h_i$ with its corresponding class prototype $p_c$ (derived from initial mean pooling), whereas query tokens are formed by concatenating the item embedding with a zero-padded vector. Regarding Attention Masks, we do not use a generic mask but rather an architectural Causal Schedule (detailed in Section 3.3). This involves a two-stage process: first, Self-Attention is applied exclusively within the Support Set to refine class prototypes; second, Cross-Attention is applied from the Query Set to the refined Support Set. Finally, the Prototype Computation is performed by taking the element-wise mean of the specialized "class-space" portion of the output embeddings for all support tokens belonging to a specific class.

---

> ### Author Response · Authors · 2025-12-03
>
> [W6] We thank the reviewer for suggesting the inclusion of stronger baselines such as SEAL and MaskGAE. As requested, we have conducted additional experiments comparing GILT against these methods and included the results in the revised PDF. On larger datasets, the fully supervised baselines leverage the abundant training signal to establish a higher performance ceiling. However, on smaller graphs, GILT actually outperforms both SEAL and MaskGAE. This validates the distinct value proposition of our framework: in low-data scenario where complex models struggle to extract stable signals without overfitting, GILT offers a superior solution with zero parameter update.
>
> [W7] We appreciate the reviewer’s suggestion to evaluate on larger and more diverse node-level datasets. We acknowledge that the Planetoid family and WikiCS are relatively small; however, our selection was driven by two specific constraints: comparability and task-specific scale coverage.
>
> First, regarding comparability, the Planetoid family and WikiCS represent the common intersection reported across the widest range of baselines, including Tuning-Based, ICL, and LLM-based methods. To ensure a direct, controlled comparison against the full spectrum of competitors without introducing confounding variables (such as differing data splits or reproduction discrepancies on less common datasets), we prioritized these standard benchmarks.
>
> Second, regarding scale, we emphasize that our evaluation as a whole covers a broad spectrum of scales. While the node classification tasks focus on standard few-shot benchmarks, our Link Prediction experiments utilized ogbl-collab (approx. 235k nodes and 1.2M edges), and our Graph Classification experiments utilized ogbg-molpcba (over 400k graphs). This confirms that GILT scales effectively to large networks. Regarding diversity, GILT is a Unified Framework designed to generalize across Node, Link, and Graph tasks. Our evaluation as a whole covers a significant diversity of domains including Citation (Cora/Citeseer), Co-purchase (Amazon Photo/Computers), Co-authorship (ogbl-collab), and Molecular Chemistry (ogbg-mol*). This confirms that the model handles diverse topologies and feature semantics (from Bag-of-Words to Chemical Fingerprints) effectively.
>
> [W8] We apologize for any confusion caused by the terminology. We clarify that the descriptor "parameter-free" was intended to characterize the GCN Aggregation step, not the entire encoder module or the downstream Transformer. We will revise the paper to avoid any confusion.
>
> [Minor]: We thank the reviewer for the detailed attention to the manuscript. Regarding the inconsistency in Table 4, we clarify that this was a formatting oversight where GILT was reported in percentages while baselines were in decimals. We will unify all metrics to the decimal format in the final revision to ensure clarity. Regarding the Typo in Equation (2), we confirm this is a notation error and will correct the vector multiplication term. Regarding the cross-task link prediction evaluation, we agree with the reviewer that reusing pre-training graphs conflates task transfer with graph reuse. To address this valid concern, we have re-conducted this experiment using strictly held-out datasets that were not seen during pre-training. The updated results, provided in the Rebuttal PDF, confirm that GILT maintains its superior zero-shot performance even on completely unseen graphs. Regarding the feature-alignment procedure (PCA) and inductive settings, we confirm that PCA statistics are computed on the full unlabeled target graph. We respectfully clarify that a "strict inductive" setting, where a model processes a single node in isolation, is fundamentally incompatible with GNNs, which inherently require the neighbors to perform message passing. Therefore, our approach follows the standard protocol in the literature. For a detailed discussion on why this unsupervised normalization does not constitute data leakage, we kindly refer the reviewer to our response to Reviewer FVan (Q2).

---

### Official Review · Reviewer_B818 · 2025-11-01

**Soundness:** 1
**Presentation:** 2
**Contribution:** 1
**Rating:** 2
**Confidence:** 5

**Summary:**

This paper proposes GILT, an LLM-free and tuning-free framework for graph foundation modeling. It introduces a token-based in-context learning mechanism that unifies node-, edge-, and graph-level tasks while effectively handling heterogeneous and numerical graph features. Experiments demonstrate that GILT achieves superior few-shot performance.

**Strengths:**

1. The paper is well-written.
2. The paper identifies critical challenges of existing GFMs which heavily rely on textual features and dataset-specific tuning.

**Weaknesses:**

1. The use of PCA for unifying heterogeneous features across graphs is not well justified. PCA only performs dimensionality reduction and does not align the semantic spaces between different graphs, making the proposed graph encoder conceptually ill-defined.

2. The novelty of the work is limited. Prior studies have already explored unifying node-, edge-, and graph-level tasks under few-shot settings, and the use of attention mechanisms for fusing support and query representations is also well-established, offering little technical innovation.

3. The experimental design appears severely biased, with problematic baseline choices and insufficient details on dataset processing.

   3.1 Table 2 compares GILT with other GFM methods. However, existing GFMs are known to perform poorly on many datasets, regardless of the reported gains in their original papers. It is crucial to include results from traditional baselines such as semi-supervised GNNs and self-supervised GNNs to better assess the true capabilities of GILT.

   3.2 Table 3(a) compares GILT with GCN and GraphSAGE on link prediction. The data split used in this setting is unclear. Do the baseline methods use the same training and test edges as GILT? If so, training a link predictor on only five edges is highly unfair, as such a small number of samples is insufficient to capture the complex dynamics of link formation. Under this extreme setting, it is unsurprising that in-context learning performs better than trained models. However, this comparison is severely biased and does not reflect the true performance of different models in realistic scenarios. Simple link prediction heuristics (e.g., common neighbors) might even outperform the chosen baselines. To properly validate the method, results under varied shot settings and with stronger baselines should be provided.

   3.3 Table 3(b) again compares GILT with GFMs for link prediction. Since GFMs are known to perform poorly, GNN baselines should also be included for a fairer comparison. Additionally, prior work [1] suggests that AUC is not an appropriate metric for link prediction; metrics such as HITS@K or MRR should be considered instead.

   3.4 Table 4 reports AUC scores for graph classification tasks, but both the baselines and the proposed method achieve nearly random performance (AUC around 0.5). Such results are insufficient to demonstrate the claimed in-context learning capability of GILT.

   3.5 Table 6 compares GILT with GFMs, but the reported results appear unusually low. For example, the 20-shot Cora accuracy of 78% is significantly below commonly reported values in the literature. More details on dataset splits and baseline are needed.

4. Overall, the experimental section would benefit from a clearer description of the experimental setup and dataset preprocessing, as well as a more comprehensive evaluation including varied shot settings and stronger, more reasonable baselines.

[1] https://arxiv.org/abs/2306.10453

**Questions:**

1. What is the rationale behind cross-task transfer of GILT? Why would node-level pretraining be beneficial for link-level or graph-level tasks?

2. Is the in-context learning model really making progress, or is it just because the baselines are too bad?

---

> ### Author Response · Authors · 2025-12-03
>
> [W1] Regarding the "ill-defined" nature of the PCA encoder, we fully agree that PCA cannot achieve semantic alignment across heterogeneous graphs. However, as detailed in our General Response, we demonstrate that strict semantic alignment is not required for effective few-shot transfer. Our results show that PCA outperforms complex learnable aligners (like FUG) because it relies on robust statistical invariants rather than overfitting to source-domain semantics. We refer the reviewer to the General Response for the detailed mechanism and the new empirical results.
>
> [W2] We thank the reviewer for accurately identifying the foundational components of our framework. We fully acknowledge that task unification and attention mechanisms are established concepts; however, we clarify that GILT represents a distinct architectural synthesis engineered to solve specific, previously unresolved challenges of GFM, where naive combinations of these components fail.
>
> First, regarding task unification, we argue that GILT introduces a fundamentally different paradigm compared to prior studies. Existing unified models (such as Prodigy, OFA) typically rely on Large Language Models to bridge feature gaps via textualization, which incurs massive computational costs. Crucially, GILT also diverges from traditional ICL approaches, which attempt to adapt to new tasks by modifying the graph topology and relying on GNN message passing to propagate prompt information. We argue that such adaptation is inherently limited by local information and high re-encoding latency. In contrast, GILT is the first framework to replace topological propagation with Transformer-based In-Context Learning. This architectural shift yields a massive experimental advantage, as demonstrated in Table 2.
>
> Second, while Attention is used in Matching Networks (CV) and TabPFN (tabular), we respectfully argue that the translation of these concepts to the domain of Heterogeneous Graphs is a non-trivial challenge that requires distinct architectural innovations. Unlike image or tabular domains where input dimensions and semantics remain consistent across samples, graph transfer learning faces extreme heterogeneity where feature spaces are disjoint and topological structures vary wildly across datasets. A naive application of standard cross-attention architectures (like TabPFN) to such data fails because the standard tabular models cannot encode the crucial topological dependencies of graph data. GILT addresses this through a novel Graph-Native Tokenization pipeline. We do not simply feed raw features into a Transformer; we engineered a specialized preprocessing stage that syntactically unifies diverse graphs into a standardized metric space. The second non-trivial innovation is our solution to the varied class problem. A fixed Foundation Model must handle tasks with arbitrary class counts without retraining a classification head. We solve this via our Asymmetric Token Formulation and Prototypical Output Interface. This design enables the first truly tuning-free architecture that can dynamically adapt to any N-way, K-shot graph task purely through forward-pass inference.
>
> Thus, GILT is not a re-application of existing tools, but a purpose-built architectural synthesis. The novelty lies in engineering this specific interface that bridges the gap between static deep learning architectures and dynamic, heterogeneous graph tasks.
>
> [W3.1] We appreciate the reviewer's suggestion to include traditional baselines like semi-supervised GCNs to better assess GILT's capabilities. We have added these comparisons. We report that fully supervised GCNs (trained from scratch on the target domain) generally achieve higher accuracy than GILT. However, we respectfully argue that this is an expected "apples-to-oranges" comparison due to the fundamental difference in problem settings: GCN is allowed to see thousands of labeled examples, update parameters via gradient descent for hundreds of epochs, and overfit specifically to the target dataset's manifold. GILT is restricted to only 5-shot examples, is forbidden from updating parameters, and must generalize using only a fixed encoder pre-trained on a disjoint domain. Therefore, we treat the fully supervised GCN as a Supervised Reference Standard, a benchmark representing the performance achievable when abundant labeled data and computational resources for gradient updates are available. The significance of GILT lies in its ability to recover a substantial fraction of this reference performance while operating in a strict 5-shot, zero-tuning scenario.

---

> ### Author Response · Authors · 2025-12-03
>
> [W3.2] We apologize for any confusion regarding the experimental setup for Link Prediction. The baseline methods did not use the same limited training edges as GILT. The GCN and GraphSAGE baselines were trained in a fully supervised manner using standard splits (e.g., 70%-10%-20%）for Planetoid and official splits for OGB. In contrast, GILT was evaluated in a strict few-shot manner, utilizing only the 5-shot support set without any gradient updates. Regarding the reviewer's suggestion to compare against stronger baselines, we have included comparisons with SEAL and MaskGAE in the revised PDF.  We observe that while these SOTA link prediction models leverage massive supervision to outperform GILT on large-scale datasets, GILT remains competitive on smaller datasets even without training, offering a distinct advantage in low-data and low-latency scenarios. Finally, regarding varied shot settings,the ablations in the appendix show that GILT’s performance saturates quickly after 5-10 shots, confirming that the 5-shot setting is a representative operating point for this metric-learning architecture.
>
> [W3.3] Regarding the choice of metrics, we fully agree with the reviewer that ranking metrics like HITS@K or MRR are generally preferred for link prediction. We emphasize that for our primary comparison with supervised baselines in Table 3(a), we strictly employed these standard ranking metrics (HITS@K). However, for Table 3(b), we adhered to AUC specifically to ensure a valid, direct comparison with the reported results of Tea-GLM, a seminal work in the GFM literature. We respectfully note that re-evaluating massive LLM-based baselines on new metrics is computationally prohibitive within the rebuttal window. Furthermore, regarding the inclusion of additional few-shot GFM baselines, we note a structural fragmentation in the current literature: the vast majority of existing few-shot GFMs are architecturally specialized for node classification or knowledge graph completion and do not natively support standard link prediction. Consequently, our comparison is necessarily restricted to the subset of foundation models that actually possess the architectural capability to perform this specific task.
>
> [W3.4] We respectfully disagree with this characterization in the context of few-shot learning on challenging benchmarks like OGB. While an AUC of 0.50 indeed represents random guessing, a score of 0.63 represents a +13% absolute improvement over random chance. In the highly constrained 5-shot task, where the model sees only ten examples to capture the complex global semantics of molecular properties in datasets like ogbg-hiv with 41,127 graphs or ogbg-pcba with 437,929 graphs, this margin indicates a significant, non-trivial learning capability.
>
> [W3.5] We respectfully clarify that this discrepancy arises from comparing two fundamentally different experimental regimes: the Standard Semi-Supervised Setting versus the Tuning-Free Few-Shot Setting. The "commonly reported values" for Cora correspond to fully trained GNNs that perform hundreds of epochs of gradient descent. In contrast, the results reported in Table 6 are strictly for GFMs operating under inference-only constraints. The baseline results listed in this table are directly sourced from the official figures reported in well-respected studies. Within this specific literature context, a 78% accuracy for a model that performs no parameter updates is highly competitive and represents a strong result.
>
> [W4] We thank the reviewer for the comprehensive summary of the experimental limitations. We acknowledge that the lack of explicit detail in the original manuscript likely contributed to the concerns regarding baseline fairness. In response, we have extensively revised the experimental section to explicitly detail the dataset preprocessing and the experiment setup. We believe these revisions directly address the reviewer’s call.

---

> ### Author Response · Authors · 2025-12-03
>
> [Q1] We thank the reviewer for this insightful question. We argue that the cross-task transfer is beneficial because GILT does not learn task-specific decision rules, but rather learns to calibrate the underlying metric space based on Feature Smoothness. Node-level pre-training on massive datasets forces the model to identify and amplify the discriminative feature patterns while suppressing stochastic noise. This process results in a refined, high-SNR metric space where entities with similar semantic content are geometrically clustered. This "denoised" geometry acts as a universal foundation that transfers directly to downstream tasks: a metric space that successfully clusters nodes by their latent identity is inherently better suited for predicting interactions (Link Prediction) or aggregating global properties (Graph Classification) than the raw, noisy feature space.
>
> [Q2] We thank the reviewer for this critical question. We clarify that the impression of "bad baselines" likely stems from the asymmetric nature of the comparison that pitching our 5-shot, zero-tuning model directly against fully supervised GNNs trained on the complete dataset. We regard these supervised baselines not as direct competitors, but only as reference standards. The fact that GILT approaches or matches this standard while utilizing little labeled data and performing no parameter updates is precisely where the progress lies.

---

### Author Response · Authors · 2025-12-03
**General Response: Addressing Semantic Inconsistency of PCA**

A shared concern among Reviewers is the semantic inconsistency of the PCA-based encoder. Since PCA aligns features based on variance rather than semantic meaning, "Dimension 0" in Graph A is semantically distinct from "Dimension 0" in Graph B. However, we provide a new perspective that GILT is not designed as a semantic feature extractor, but rather as a **Universal Spectral Denoiser**.

Our framework relies on a Prototypical Head, which computes predictions based on the distance between Query and Support tokens. The performance of such metric-based learners depends critically on the Signal-to-Noise Ratio of the embedding space. We leverage the well-established observation from Spectral Graph Theory that real-world graphs exhibit an "L-shaped" spectral distribution. Let the magnitude of the $i$-th PCA component be denoted by $|x_i|$. We model this spectral profile as a superposition of a latent structural signal $s_i$ and stochastic noise $n_i$, such that $|x_i| \approx |s_i| + |n_i|$. Consistent with the heavy-tailed degree distributions of scale-free networks, we assume the structural signal energy follows a Power Law decay, $\mathbb{E}[|s_i|^2] \propto i^{-\alpha}$ (where $\alpha > 1$), while the stochastic noise spectrum approximates a uniform White Noise floor, $\mathbb{E}[|n_i|^2] = \sigma^2$. GILT exploits this invariant shape through a two-stage filtering process: spatial consensus (MHA) followed by spectral gating (FFN).

The MHA mechanism functions as a **Geometric Consensus Filter**. Crucially, the attention mechanism relies on the dot product ($Attention(Q, K) \propto Q \cdot K^T$), an operation that is inherently semantically insensitive; it measures the geometric alignment rather than matching semantic features. By attending to the Support set, the Query effectively projects its noisy spectrum onto a stable Class Centroid. This process smooths out idiosyncratic noise spikes and refines the spectral estimate before it reaches the FFN.

The FFN then functions as an **Adaptive Spectral Gate**. First, the linear transformation learns to attenuate high-frequency indices in alignment with the Power Law decay. Instead of targeting specific semantic features, the network effectively approximates the broad statistical trend of the data, maintaining high sensitivity to the information-rich lower indices while progressively dampening the higher indices where the signal probability diminishes. Subsequently, the ReLU activation functions as a magnitude-based gate, establishing a cut-off threshold to suppress the background noise floor.

This spectral denoising mechanism is consistent with the ICL capabilities of the model. Successful ICL requires the model to identify correlations between the Support and Query sets. By functioning as a Spectral Denoiser, the pipeline cleans the metric space, ensuring that the Attention in the ICL module operates on robust structural invariants rather than noise.

To empirically validate the feasibility of our PCA-based approach, we compare PCA against Random Feature, Random Projection and the SOTA learnable aligner FUG under the same 5-shot setting in Table 2. We find that replacing features with Random Noise results in a performance collapse to near-random guessing, confirming that the input features contain critical structural information. Furthermore, our results demonstrate that PCA significantly outperforms Random Projection, confirming that the variance-sorting property is necessary to distinguish structural signal from stochastic noise. Moreover, our fixed PCA approach surpasses FUG framework even when FUG is trained end-to-end. This comparison highlights that FUG still attempts to approximate a statistical alignment via gradient descent. However, in the few-shot regime, this optimization is highly unstable and often results in a distorted feature space. In contrast, PCA provides the analytical, closed-form solution to the variance-maximization problem.

| Dataset | PCA | Random Feature | Random Projection | FUG|
| :--- | :--- | :--- | :--- |:--- |
| Cora | $70.58 \pm 2.75$ | $39.00 \pm 18.6$ | $57.71 \pm 4.25$ | $63.30 \pm 1.25$ |
| Citeseer | $61.44 \pm 0.57$ | $33.60 \pm 2.96$ | $37.97 \pm 3.66$ | $41.88 \pm 4.41$ |
| Pubmed | $64.94 \pm 7.48$ | $48.23 \pm 3.78$ | $63.24 \pm 2.86$ | $65.23 \pm 6.44$|
| WikiCS | $69.40 \pm 2.02$ | $19.77 \pm 4.42$ | $52.56 \pm 5.17$| $60.60 \pm 4.24$|

---

### Meta-Review · Area_Chair_tGiE · 2026-01-01

**Summary:**

This paper proposes a graph foundation model (GFM) that is LLM-free and tuning-free. It leverages the in-context learning capability of the model by performing graph native tokenization that separately captures the feature and structural information. A two-stage transformer architecture is proposed that handles cross attention over support tokens, and operates a prototypical head for jointly handling node-, link- and graph-level tasks. Experiments on various graph learning tasks under different few-shot settings have been performed against different baseline families, showing promising efficiency gains due to the tuning-free property.

**Reviewer Concerns:**

# Reviewer B818:

### Addressed:
* The concerns on experimental setup and biased evaluation should have been partially addressed. e.g., the data split for supervised baselines; the choice and scale of AUC, etc.
* The response on the benefits of cross-task transfer makes intuitive sense. Although it may also be true that different tasks can be susceptible to different types of noises.

### Unaddressed:
* The concern on PCA in terms of feature space alignment is valid and remains outstanding. Although the authors have provided common response interpreting PCA as denoiser, it is arguable that denoising and alignment are two separate problems and feature space alignment does appear to be a fundamental requirement for any training-free downstream model to perform reasonably. In addition, the interpretation on the denoiser relies on strong assumptions on the input graph (e.g., its spectrum and noise pattern) and it is unlikely that the inputs to a foundation model would be constrained within such a narrow class.
* The concerns on limited performance gain / non-informative baselines still partially remain. The classic GNN baselines (that requires supervised training) perform much better on many of the selected baselines. It is true that requiring supervised training is their fundamental drawback, but it is not clear why such training overhead would be a major issue on graphs of such small scales. In other words, the benefits of the proposed tuning-free GFM may not be practically significant on those small graphs. In addition, I agree that "simple link prediction heuristics (e.g., common neighbors) might even outperform the chosen baselines". Comparison with those parameter- & training-free methods would be interesting.


# Reviewer h6fb

### Addressed:
* The novelty and distinction from existing methods have been well discussed in the response
* The authors have incorporated suggested changes on clarifying linear GNN models, additional baselines like SEAL and MaskGAE etc.
* The authors have clarified design choices such as normalization, etc.

### Unaddressed:
* The evaluation is in general based on small scale benchmarks.
* Benefits over additionally included baselines (SEAL, MaskGAE in Table 3a) are not clear. The few-shot setting still lags behind the supervised model by a large margin considering the average performance.


# Reviewer uE5v

### Addressed:
* Empirical evidence on PCA vs random d-dimensional features

### Unaddressed:
* Lack of justification on the feature space alignment through PCA. See summary above.


# Reviewer FVan

### Addressed:
* It is reasonable that accuracy should not be a single metric when evaluating foundation model performance. The capability of GILT to handle different levels of tasks is indeed a unique advantage.
* The conceptual clarification on the ICL mechanism makes sense.

### Unaddressed:
* The concern on PCA. See above.

**Reviewer Scores:**

The authors have provided thorough responses in the rebuttal. I believe all reviewers would have their concerns partially addressed. So I would expect some reviewers to increase their original ratings.

However, some concerns still remain and I encourage the authors to clarify them in future revisions.

---

### Decision · Program_Chairs · 2026-01-26

Reject